# Experimental Study on the Negative Skin Friction of Piles in Collapsible Loess

**Qing Chai [1], Tianlei Chen [1], Zuoyong Li [2], Danyi Shen [2,3,*] and Chuangzhou Wu [2]**

[1]  Lanzhou Engineering & Research Institute of Nonferrous Metallurgy Co., Ltd., No. 168 South Tianshui Road, Lanzhou 730000, China; chentl2023@163.com (T.C.)

[2]  Institute of Port, Coastal and Offshore Engineering, Ocean College, Zhejiang University, Zhoushan 316021, China

[3]  Institute of Geotechnical Engineering, College of Civil Engineering and Architecture, Zhejiang University, Hangzhou 310058, China

*  Correspondence: shendanyi@zju.edu.cn

**Abstract:** The collapsible loess is widely distributed in western China. The special structure and water sensitivity of loess lead to the complex negative skin friction mechanism in pile foundations. Previous studies mainly focused on the negative skin friction of pile foundations and treatment measures, such as casing and coating methods. However, few studies have focused on the influence of the negative skin friction on the settlement and bearing capacity of piles in collapsible loess, especially environmentally friendly methods that can reduce the negative skin friction. In this study, a series of non-immersion and immersion experiments was conducted to investigate the settlement, axial force, and side friction resistance of piles in loess soil under controlled conditions. The results showed that under the non-immersion condition, the settlement of model piles increased with the increasing pile top load. The axial force gradually decreased along the pile length for piles without casing. The axial force attenuation of the casing section of casing piles was almost negligible due to the isolating frictional resistance effect of casing. The settlement of each soil layer increased with the increase in immersion time, and the process was divided into an initial gradual stage, rapid drop stage, and later gradual stage. Both negative and positive skin friction increased with the increasing immersion time and pile top load, and there was a neutral point. The maximum axial force of piles without casing exceeded the peak load at the pile top. The presence of steel casing reduced the failure of pile foundation in collapsible loess. The research results of this paper provide theoretical support for the application of piles in loess areas.

**Keywords:** collapsible loess; pile; casing; axial force; negative skin friction

## 1. Introduction

Collapsible loess is a type of soil that is prone to softening and collapsing when it comes into contact with water [1,2]. The collapsible loess is widely distributed in western China, and leads to serious engineering problems, such as loss of bearing capacity, differential settlement, and even structural failure. In the case of pile foundations, the negative skin friction caused by the softening and collapsing of the loess under immersion conditions can significantly reduce the bearing capacity of the piles and result in uneven settlement of the foundation [3,4]. Therefore, it is essential to study the influence of the negative skin friction on the settlement and bearing capacity of pile foundations in collapsible loess areas.

Extensive research on negative skin friction and the neutral depth of pile foundations from different perspectives including theoretical research, model test and numerical simulation were conducted in recent years [5–7]. Poulos et al. [8] proposed a negative skin friction calculation method using the Mindlin solution based on elastic theory, and combined it with Terzaghi's one-dimensional consolidation theory to obtain the relationship between negative skin friction and time. Gao et al. [9] presented a pile–soil load transfer

function that can consider both the nonlinearity and the ultimate shear strength of soil to calculate the negative skin friction of piles. Kim et al. [10] compared the development of negative skin friction on a steel pile and a concrete pile based on several laboratory model tests. Zhao et al. [11] analyzed the change and distribution law of negative skin friction of pile foundations in collapsible loess areas according to model tests, and found that both negative skin friction and positive skin friction existed on the pile surface. Liu et al. [12] established a two-dimensional axisymmetric model and analyzed the neutral plane and the distribution of negative skin friction along the pile length. Chiou and Wei [4] investigated the influence of pile top load on the development of negative skin friction in friction single piles and friction-end-bearing single piles based on ABAQUS software. Despite these prior studies, the special structure and water sensitivity of loess lead to the complex negative skin friction mechanism in pile foundations. It is necessary to analyze the impact of water immersion on negative skin friction through model tests [13].

Several treatment measures, such as the casing method, coating method, and new types of piles are widely used in collapsible loess to reduce the negative skin friction of pile foundations [14,15]. Bjerrum et al. [16] developed methods of applying bitumen to piles to reduce negative skin friction, and take precautions to prevent the bitumen from being destroyed as the piles pass through the filler. Yang [17] presented a new type of steel tubular composite pile and analyzed its bearing characteristics and the rationality in collapsible loess. Dong et al. [18] proposed a caterpillar pile that can both reduce negative skin friction and increase positive skin friction in collapsible loess. However, the materials used in the pile-coating process contain harmful chemicals that may contaminate groundwater. New types of piles that can reduce the negative skin friction need to be further studied. Moreover, research of piles mainly focusing on the reduction of negative skin friction, the change law of the pile axial force, and the pile side friction resistance need to be explored.

Consequently, the objectives of the present study are: (i) To conduct a series of non-immersion and immersion experiments of piles in loess soil under controlled conditions. (ii) To investigate the axial force and side friction resistance of both concrete piles and steel piles with and without casing under pile top load and non-immersion conditions. (iii) To analyze the settlement of loess soil and bearing characteristics of piles with or without additional surface load under immersion conditions. This study provides theoretical support for the application of piles in collapsible loess foundations.

## 2. Methodology

### 2.1. Similarity Ratio Design

The similarity ratio for the model experiments were determined based on the gravity similarity laws and dimensional analysis methods [13,19]. Table 1 lists the similarity relations and similarity coefficients of the physical quantities of the model. All of the elastic modulus, density and linear displacement are controlled by model design [13].

**Table 1.** Similarity relation.

| Physical Quantities | Similarity Relation | Similarity Ratio |
|---|---|---|
| Strain, $\varepsilon$ | $C_\varepsilon = 1.0$ | 1 |
| Stress, $\sigma$ | $C_\sigma = C_E$ | 1/3.6 |
| Elastic modulus, $E$ | $C_E$ | 1/3.6 |
| Poission's ratio, $\mu$ | $C_\mu = 1.0$ | 1 |
| Density, $\rho$ | $C_\rho$ | 1 |
| Length, $l$ | $C_l$ | 1/10 |
| Area, $S$ | $C_S = C_l^2$ | 1/100 |
| Linear displacement, $X$ | $C_X = C_l$ | 1/10 |
| Concentrated force, $p$ | $C_p = C_E C_l^2$ | 1/360 |
| Surface load, $q$ | $C_q = C_E$ | 1/3.6 |

### 2.2. Synthetic Collapsible Loess

The collapsibility of loess is closely related to its structural properties. In order to ensure the performance of model tests of synthetic collapsible loess were similar to that of natural collapsible loess, a mixture of five materials, namely sand, quartz powder, bentonite, gypsum, and industrial salt were used (Figure 1). The mass ratio of five materials in synthetic collapsible loess were 0.25:0.3:0.3:0.1:0.05. Sand and quartz powder were used as weighting materials, bentonite as a binding material, and gypsum and industrial salt as collapsible and sensitive materials.

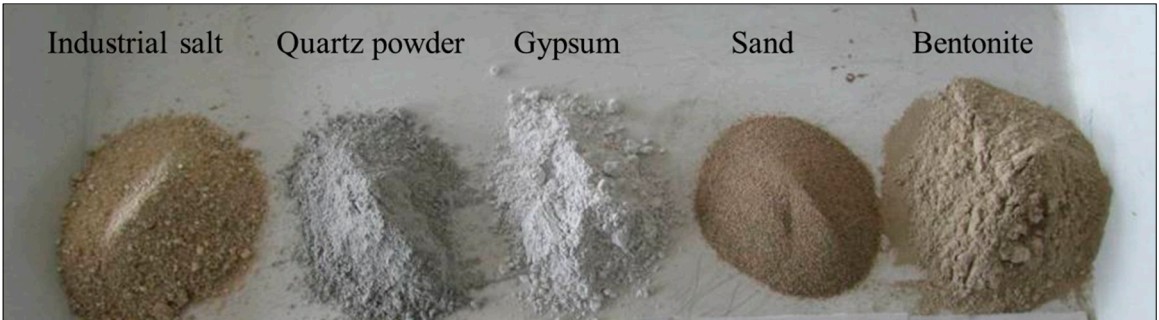

**Figure 1.** Materials of synthetic collapsible loess.

The free-drop method was used to simulate the formation of loess. First, a 1 mm sieve was placed 200–400 mm above the ring cutter, and the mixed materials were placed on the sieve. Then, gently shaking of the sieve to allowed the mixture to fall freely until the height of the mixture was higher than that of the ring cutter. The surface of the sample was leveled and lightly pressed. After that, the sample was moistened to obtain synthetic collapsible loess with different moisture contents. Finally, collapsible loess samples were placed in a constant temperature box at 50 °C for 24 h to simulate the dry and hot environments during the formation of loess.

The synthetic collapsible loess obtained had large pores and a porous loose structure, which were similar to that of natural collapsible loess. Through compaction and liquid limit tests, the maximum dry density of the synthetic collapsible loess was 1.73 g/cm$^3$, with an optimum moisture content of 16.1%, a liquid limit of 26.7%, and a plastic limit of 15.5% (Table 2). Based on direct shear and consolidation tests, the cohesive force of the synthetic collapsible loess was determined to be 64.24 kPa, the internal friction angle was 25.45°, the compression coefficient was 0.43 MPa$^{-1}$, and the compression modulus was 4.75 MPa (Table 3). By comparing the physical and mechanical properties of the natural collapsible loess, the indexes of synthetic collapsible loess were similar to those of natural loess. In addition, collapsible tests were carried out under vertical pressures of 50 kPa, 100 kPa, 200 kPa, 300 kPa, and 400 kPa. The collapsibility coefficient of synthetic collapsible loess was larger than 0.07, which belonged to strong collapsible loess. This result further confirms that the synthetic collapsibility loess is reasonable in terms of material selection and ratio design.

**Table 2.** Physical properties of synthetic collapsible loess.

| $\rho_d$ (g/cm$^3$) | $G_s$ | $e$ | $W_l$ (%) | $W_p$ (%) | PI | $w$ (%) | $\rho_{dmax}$ (g/cm$^3$) |
|---|---|---|---|---|---|---|---|
| 1.30 | 2.66 | 1.04 | 26.7 | 15.5 | 11.1 | 16.1 | 1.73 |

Note: $\rho_d$ is dry density, $G_s$ is specific gravity, $e$ is void ratio, $W_l$ is liquid limit, $W_p$ is plastic limit, PI is plasticity index, $w$ is optimal water content, $\rho_{dmax}$ is maximum dry density.

**Table 3.** Mechanical properties of synthetic collapsible loess.

| $c$ (kPa) | $\varphi$ (°) | $E$ (MPa$^{-1}$) | $E_s$ (MPa) |
|---|---|---|---|
| 64.24 | 25.45 | 0.43 | 4.75 |

Note: $c$ is cohesive force, $\varphi$ is internal friction angle, $E$ is compression coefficient, $E_s$ is compression modulus.

### 2.3. Model Piles

As shown in Figure 2, the experiment included a total of 8 model piles. The length and the radius of model piles were 3240 mm and 108 mm, respectively (Table 4). Piles 1, 2, 5, and 6 were made of fine aggregate concrete, and piles 3, 4, 7, and 8 were made of steel. Piles 5, 6, 7, and 8 were externally equipped with casing. The casing was made of steel pipe with a length of 2000 mm and a radius of 158 mm.

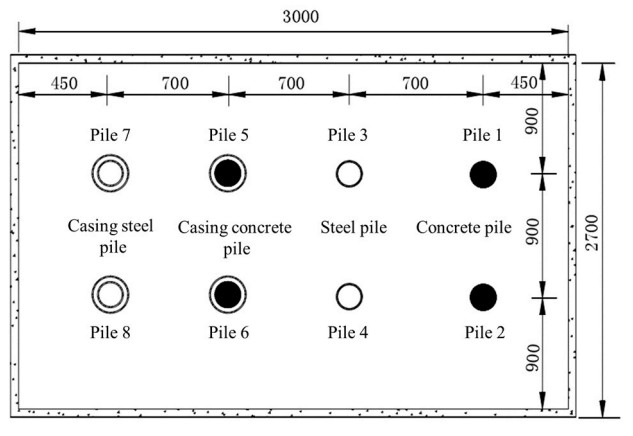

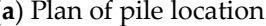

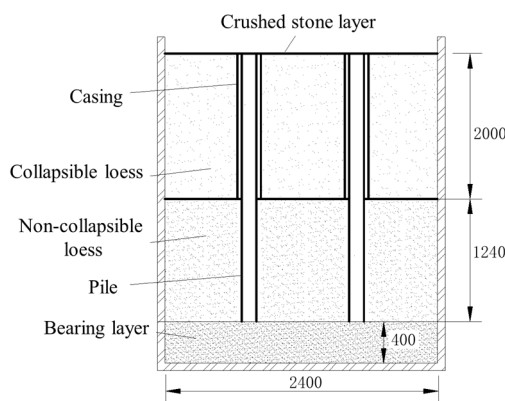

(**a**) Plan of pile location　　　　　　　(**b**) Profile of pile location

**Figure 2.** Pile location.

**Table 4.** Detail information of model piles.

| Number of Piles | | 1 | 2 | 3 | 4 | 5 | 6 | 7 | 8 |
|---|---|---|---|---|---|---|---|---|---|
| Type | | concrete | concrete | steel | steel | concrete | concrete | steel | steel |
| Diameter (mm) | | 108 | 108 | 108 | 108 | 108 | 108 | 108 | 108 |
| Length (mm) | | 3240 | 3240 | 3240 | 3240 | 3240 | 3240 | 3240 | 3240 |
| Casing | Type | / | / | / | / | steel | steel | steel | steel |
| | Length (mm) | / | / | / | / | 2000 | 2000 | 2000 | 2000 |
| | Diameter(mm) | / | / | / | / | 158 | 158 | 158 | 158 |

Strain gauges were symmetrically applied at intervals along the longitudinal direction of the model pile to obtain the axial force of the pile. Strain gauges were placed every 30 cm, and the first strain gauge was 24 cm away from the pile bottom. Each side of the model pile had a total of 12 strain gauges. The two sides were numbered sequentially from 1-1 to 1-12 and 2-1 to 2-12. Moreover, for piles with casing, strain gauges were attached to the top layer of the casing every 20 cm, with one every 30 cm below. A total of 8 strain gauges were symmetrically placed on both sides, numbered sequentially from 1-1 to 1-8 and 2-1 to 2-8.

A total of 16 loading sensors were installed at both the top and bottom of piles to measure the load they were bearing. In addition, 17 layered settlement meters were placed at $-670$ mm, $-1340$ mm, $-2010$ mm, $-2630$ mm, $-3250$ mm of the surface layer around each pile, so as to obtain the real-time settlement of soil layers and the relation curve between the load of the pile top and the immersion time, which was helpful to obtain the location of the neutral point. The locations of settlement meters are shown in Figure 3.

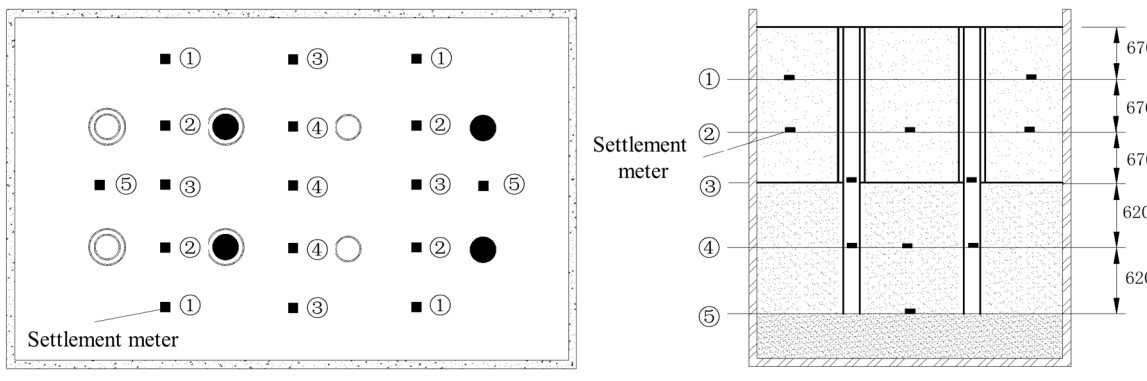

(**a**) Plan of settlement meters          (**b**) Profile of settlement meters

**Figure 3.** Location of settlement meters.

### 2.4. Experimental Design and Procedures

The experiment was conducted in the geotechnical laboratory of Lanzhou Jiaotong University. The size of the experimental model tank was 3 m × 2.7 m × 3.7 m (length × width × height), and the model tank adopted masonry structure. To ensure the impermeability of the model tank and ensure the immersion test ran smoothly, plastic films were placed on the bottom and side walls of the model tank as boundary treatment (Figure 4).

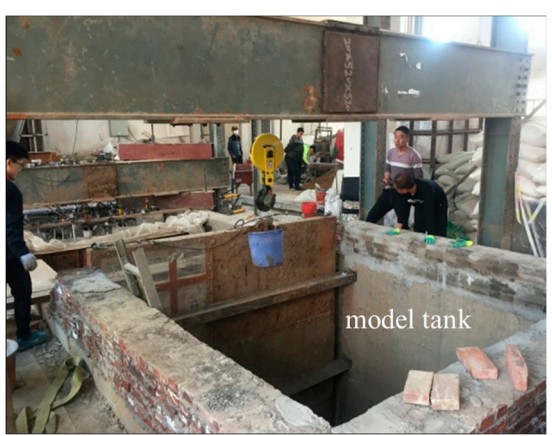
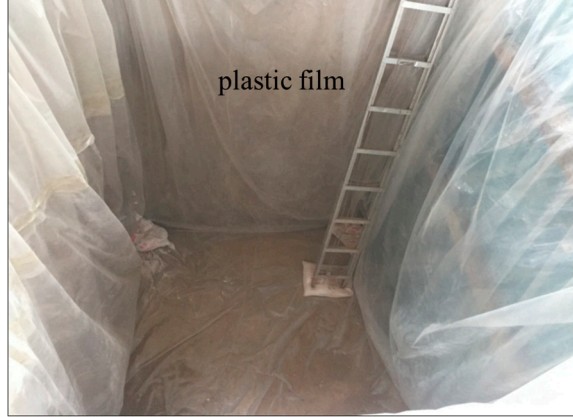

**Figure 4.** Experimental model tank.

The main experimental procedures are set out below:

(1)   At the bottom of the model tank, a 400 mm-thick sand bearing layer was filled. An electric compactor was used to slightly compress the sand every 110 mm to reach a density of 1.7 g/cm$^3$. The prefabricated model piles were suspended in the design position using ropes. The verticality of each pile was ensured by using the heavy hammer suspension method. A layer of mixed sand and loess with a thickness of 1240 mm was filled, and followed by a layer of synthetic collapsible loess with a thickness of 2000 mm (Figure 2b). The compaction degree was controlled of 0.95 by using a ring cutter method. In addition, a 100 mm thick crushed stone was applied on the surface of synthetic collapsible loess to ensure the uniform penetration of water during immersion.

(2)   A slow load maintenance method was used to load piles 2, 4, 6, and 8. The loading system employed a reaction beam and jacks, and the test load was gradually loaded. Each stage was loaded with 1.0 kN.

(3)   After applying the load, the experiment started by gradually flooding with water. The impact of adding water on the settlement of each piles and the settlement of the soil itself were observed by settlement meters. The settlement at the pile top was

measured at 5, 15, and 30 min after water immersion, and then measured every 30 min after accumulating 1.0 h. The measured data was transferred to the computer.

(4) The next stage of load was applied after the previous stage of load had reached a relatively stable state (the settlement was less than 0.1 mm per hour and occurred twice in succession). The load was increased until the lateral soil of the pile was damaged, and then the load was gradually unloaded until it reached zero.

## 3. Results and Discussion

### 3.1. Bearing Characteristics of Model Piles (Non-Immersion Condition)

Figure 5 is the relationship between load and settlement of models piles 2, 4, 6, and 8 under the non-immersion condition. During the initial phase of loading, the vertical settlement of pile top increased almost linearly with the increasing load. The settlement rate of the top of model piles 2, 4, 6, and 8 started to increase when the load was added to about 10 kN, 6 kN, 8 kN, and 5 kN, respectively. The ultimate bearing capacity of model piles 2, 4, 6, and 8 were 16 kN, 10 kN, 13 kN, and 8 kN, respectively. The experimental results showed that the bearing capacity of concrete piles without casing was the highest (pile 2), which was 3 kN larger than that of the casing concrete pile (pile 6). The bearing capacity ratios of model piles 2 and 6 were 81.3%. This result indicated that the existence of casing isolated the friction resistance, but the bearing capacity of piles still met the requirements when considering safety factors. The ultimate bearing capacity of the steel pile (pile 4) without casing was 6 kN smaller than that of concrete piles without casing (pile 2) because the friction effect between steel and soil was lower than that between concrete and soil. Additionally, the ultimate bearing capacity of the casing steel pile (pile 8) was the smallest.

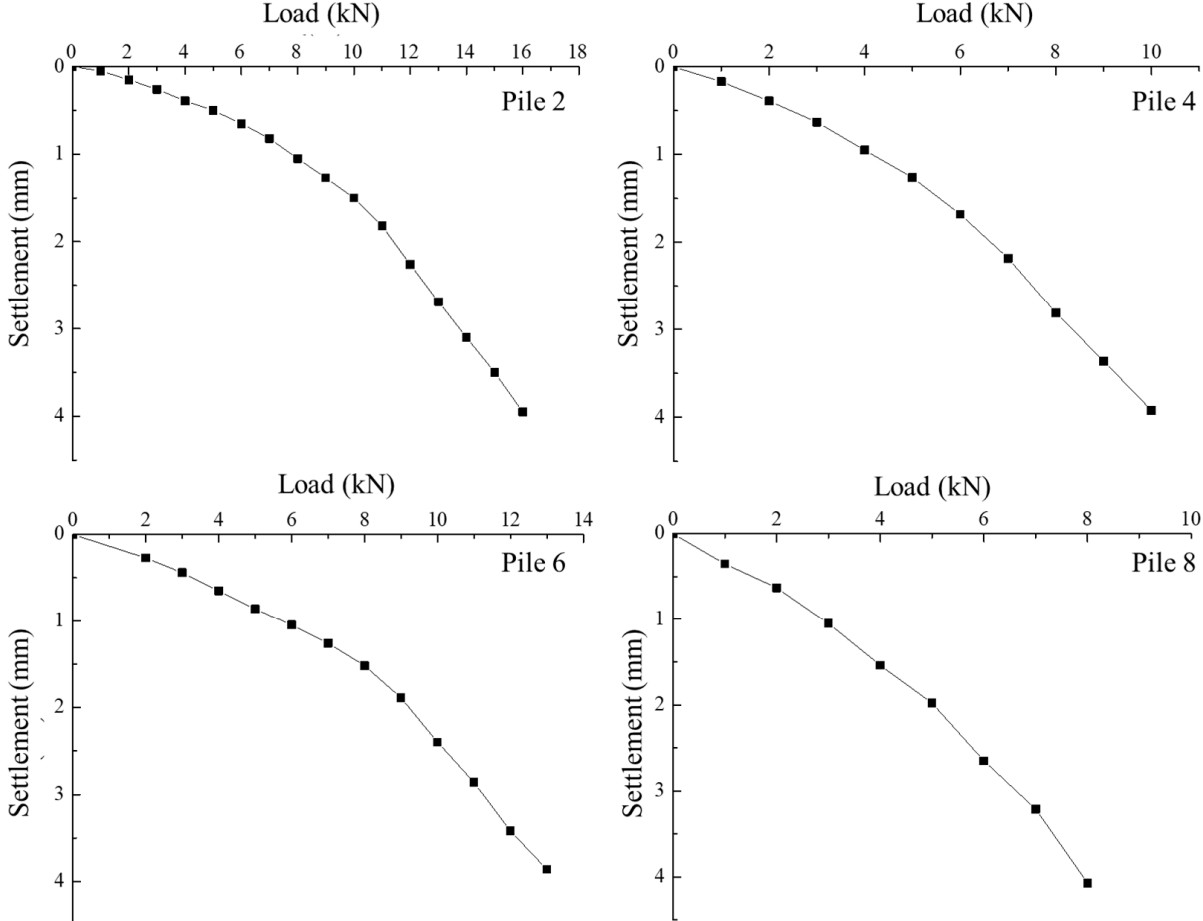

**Figure 5.** Settlement of pile top for piles 2, 4, 6, and 8.

The axial force curves for model piles 2, 4, 6, and 8 were calculated based on the strain of the piles, as shown in Figure 6. The axial force in the pile body gradually decreased along the pile length under the pile top load. The upper part of the piles bore the majority of the axial force, which reflected the characteristics of the friction pile. For the piles without casing (piles 2 and 4), the axial force of the pile was distributed relatively evenly along the pile at the beginning of loading. With the increase in load, the majority of the axial force was concentrated in the upper part of piles and the axial force transmitted to the pile end was small. For casing piles (pile 6 and 8), the axial force was transmitted to the lower non-collapsible soil layer quickly because the upper synthetic collapsible loess layer was loose. The frictional resistance exerted by the lower soil layer was larger than that of the upper synthetic collapsible loess layer. Additionally, the axial force attenuation in the casing part of the casing piles was almost zero, indicating that the isolating frictional resistance effect of the casing was significant [20].

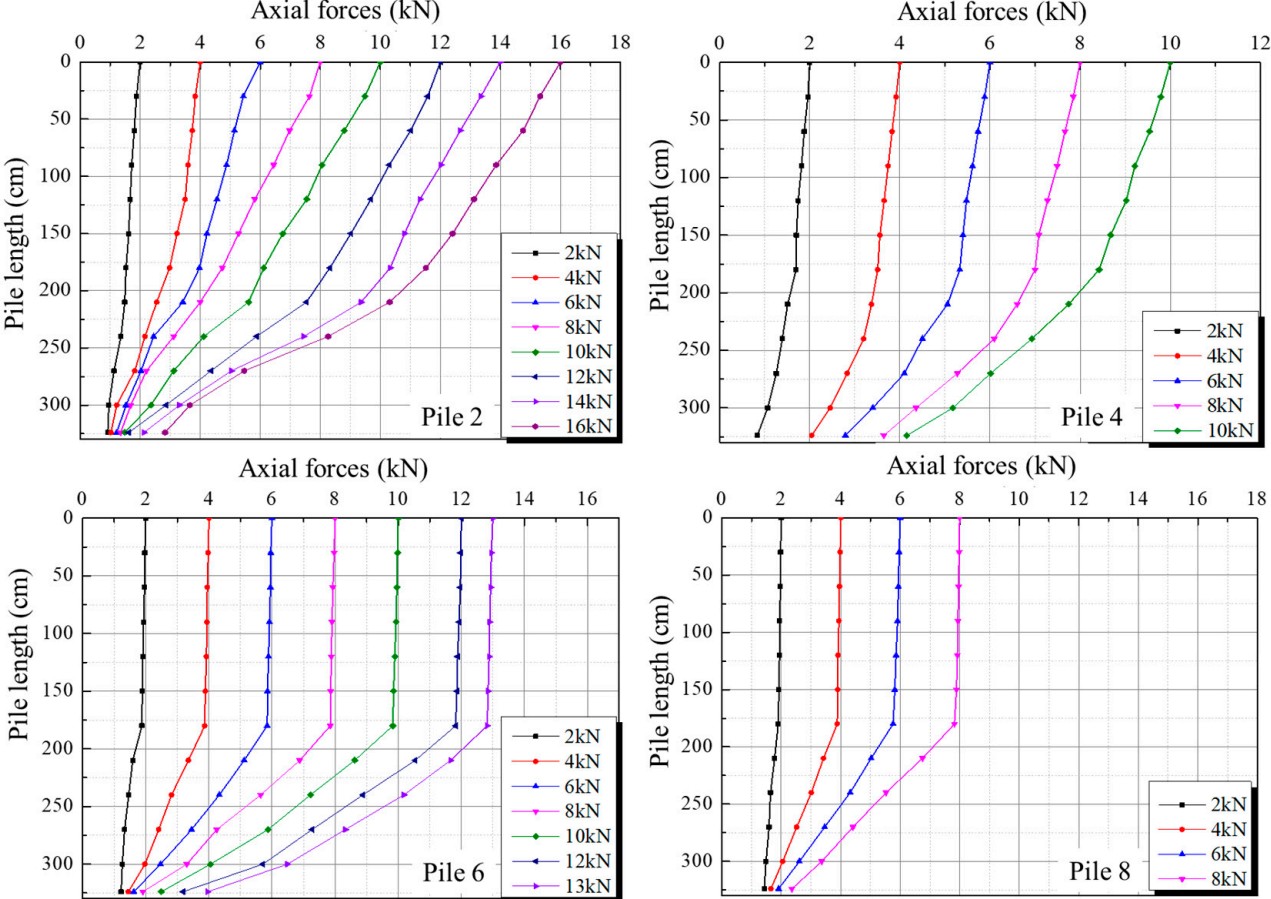

**Figure 6.** Axial force for piles 2, 4, 6, and 8 under non-immersion condition.

Figure 7 illustrates the variation in side frictional resistance along the pile for different loading conditions. The distribution of side frictional resistance along the pile was generally uniform when the load was 2 kN. Piles 4 and 6 exhibited a gradual increase in side frictional resistance along the pile length, with the maximum value occurring near the pile end. Conversely, piles 2 and 8 showed an initial increase followed by a decrease in side frictional resistance along the pile length. Furthermore, the side frictional resistance curve for piles without casing was divided into two segments: 0–1750 mm and 1750–3240 mm. In the first segment, the side frictional resistance remained relatively constant with depth but increased with the pile top load. In the second segment, the side frictional resistance changed nonlinearly and increased with increasing loading.

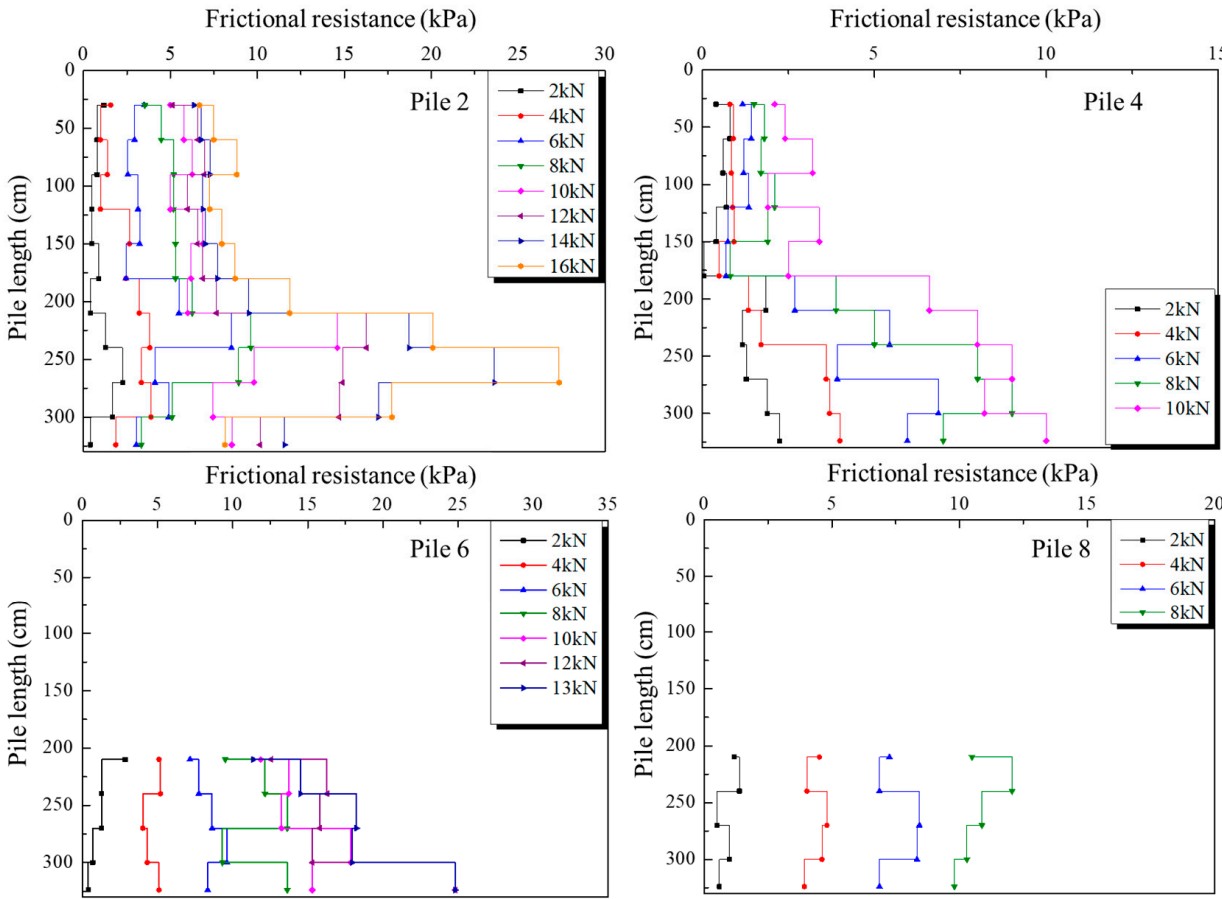

**Figure 7.** Side frictional resistance for piles 2, 4, 6, and 8.

### 3.2. Layered Settlement of Soils (Water Immersion Condition)

The relationship between the layered settlement of soils and immersion time is shown in Figure 8. During the experiment, the soil was immersed for 14 days until all layers settled and stabilized. The settlement increased as the immersion time extended, and the sedimentation mode of each soil layer was similar [11]. The settlement of each soil layer first increased rapidly and then grew steadily due to the high collapsibility of loess in the early stage of immersion. The change in settlement was divided into the following 3 stages:

(1) Initial gentle stage: during the initial 50 h after immersion, the upper synthetic collapsible loess layer experienced gradual settlement, while the middle and lower non-collapsible soil layers remained almost stable with minimal settlement.

(2) Rapid drop stage: With the infiltration of water from the shallow to deep parts of the soil layer, a steep drop period of settlement deformation occurred in different parts along the depth of soil layers. The settlement of the upper synthetic collapsible loess layer developed rapidly, and the settlement amount of the middle and lower non-collapsible soil layers also increased gradually. After about 150 h of immersion, the cumulative settlement amount at each soil layer reached almost 70% of the total settlement amount. This result indicated that the settlement of loess became stable and the lateral friction of piles tended to be fully exerted.

(3) Later gentle stage: after 150 h of immersion, the settlement rate gradually decreased, and the settlement–immersion time curve gradually flattened until the settlement reached relative stability.

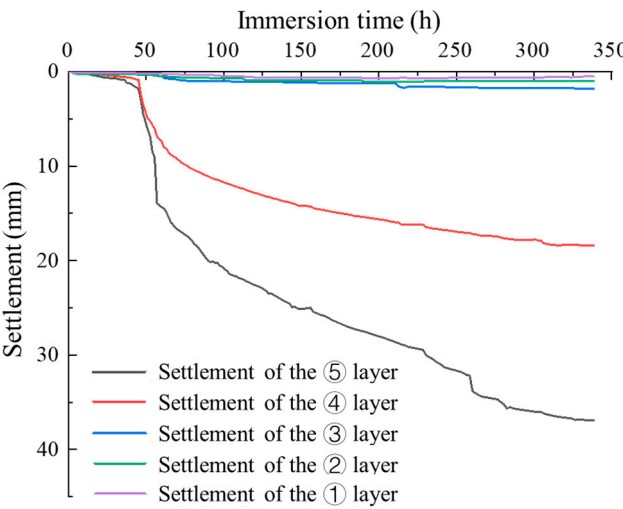

**Figure 8.** The layered settlement of soils.

### 3.3. Bearing Characteristics of Model Piles (Water Immersion Condition)

3.3.1. Without Additional Surface Loading

The axial force curves for model piles 1, 3, 5, and 7 under immersion condition without additional loads are shown in Figure 9. The immersion test lasted for 14 days, and the axial force curve exhibited a "D-shape". The axial force increased gradually after 50 h immersion, and then decreased due to the influence of negative skin friction. All model piles were in a tension state, and the peak axial forces of model piles 1, 3, 5, and 7 were 5.3 kN, 4.1 kN, 4.7 kN, and 4.3 kN, respectively.

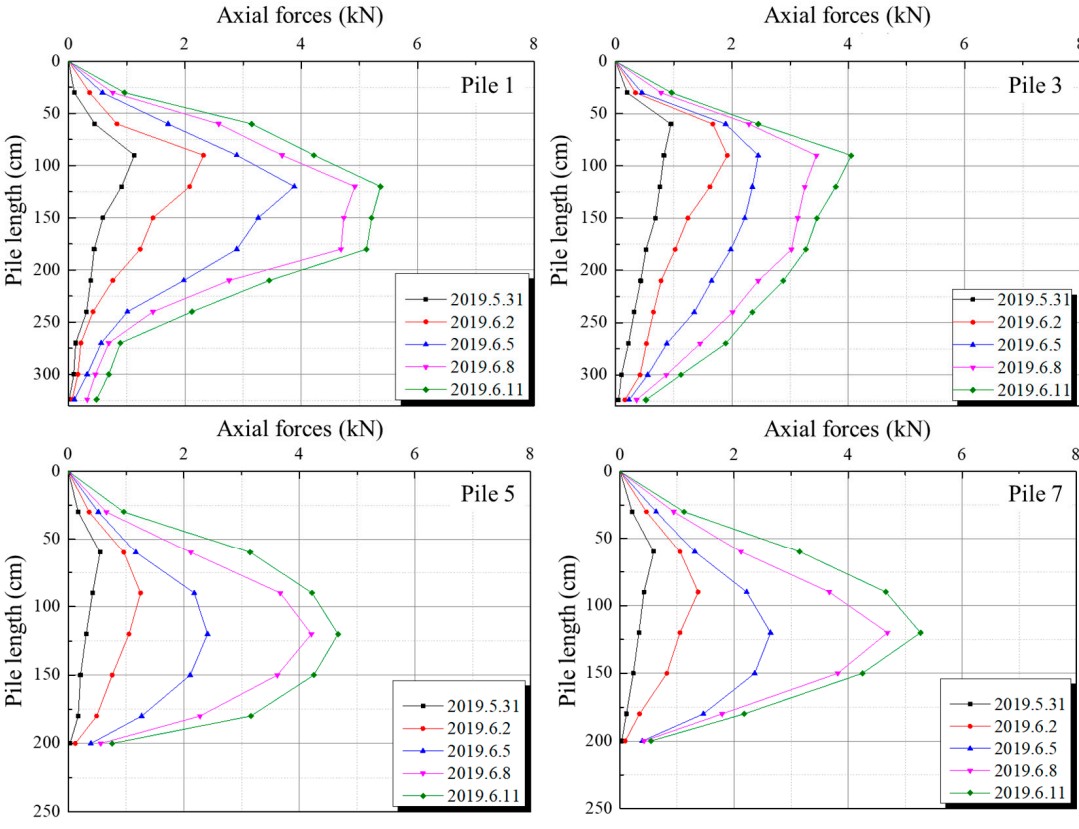

**Figure 9.** Axial force for piles 1, 3, 5, and 7.

Under the condition of water immersion, the side frictional resistance consisted of both positive skin friction and negative skin friction, as shown in Figure 10. After the model tank was immersed in water, the negative skin friction of model piles initially increased and then decreased until it reached zero. This resulted in the appearance of a neutral point, after which the positive skin friction continued to increase. The location of the neutral point varied among model piles, with piles 1, 3, 5, and 7 having neutral points at approximately 900 mm, 900 mm, 1200 mm, and 1200 mm, respectively. As the immersion time increased, both the negative skin friction and positive skin friction increased significantly. The peak value of negative skin friction of model piles reached −22 kPa, −16 kPa, −15 kPa, −14 kPa, respectively.

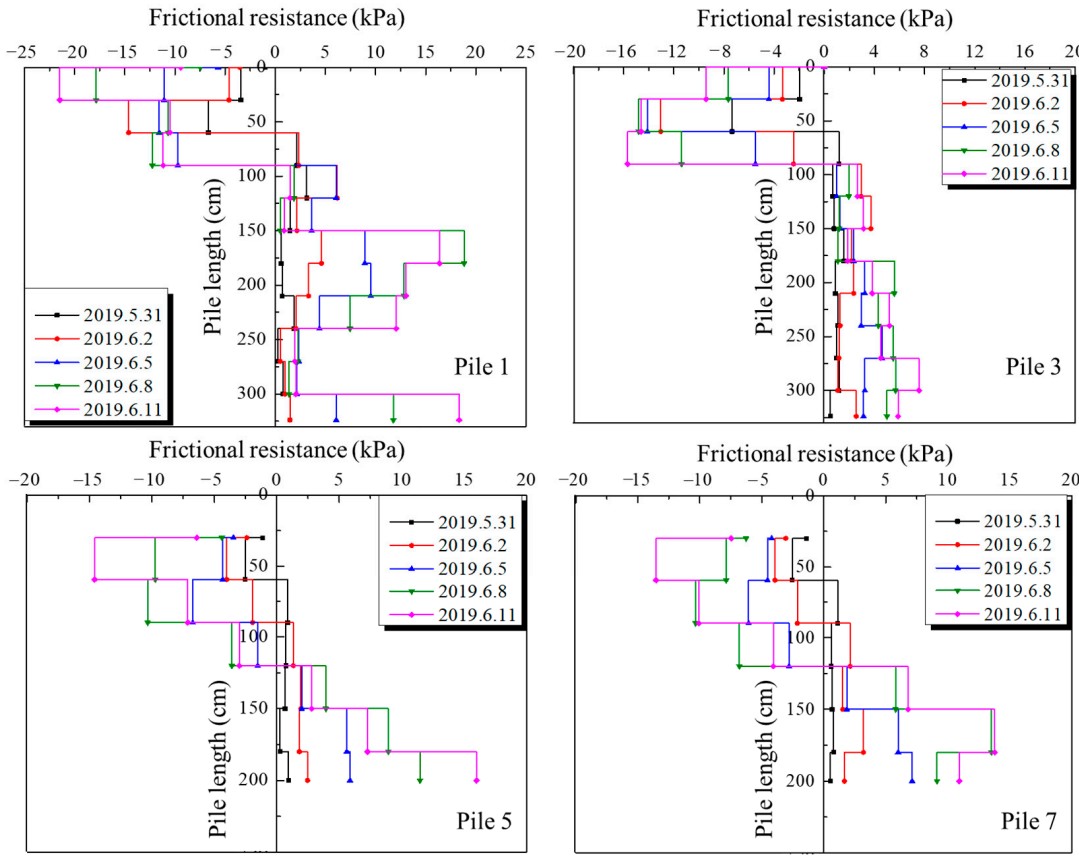

**Figure 10.** Side frictional resistance for piles 1, 3, 5, and 7.

3.3.2. With Additional Surface Loading

Figure 11 is the relationship between load and settlement of model piles 2, 4, 6, and 8 under the immersion condition. Prior to immersion, the applied loads of the four model piles were 16 kN, 10 kN, 13 kN, and 8 kN, respectively, with corresponding settlements of 3.95 mm, 3.92 mm, 3.86 mm, and 4.07 mm, respectively. After a 14-day immersion period, the settlements of model piles 2, 4, 6, and 8 increased to 8.36 mm, 10.21 mm, 9.64 mm, and 12.07 mm, respectively. The model piles were then loaded step-by-step in a saturated state. The vertical settlement of the pile top increased slowly with increasing load and then entered a stage of significant drop. The values corresponding to the sudden change point of settlement were taken as the ultimate carrying capacity of model piles. Therefore, the ultimate carrying capacities of model piles 2, 4, 6, and 8 were determined to be 34 kN, 28 kN, 30 kN, and 24 kN, respectively. Furthermore, it was found that the ultimate bearing capacity of a pile with a casing was 88.2% of that without a casing for concrete piles and 85.7% for steel piles. This indicated that the casing had a significant isolation effect on negative skin friction.

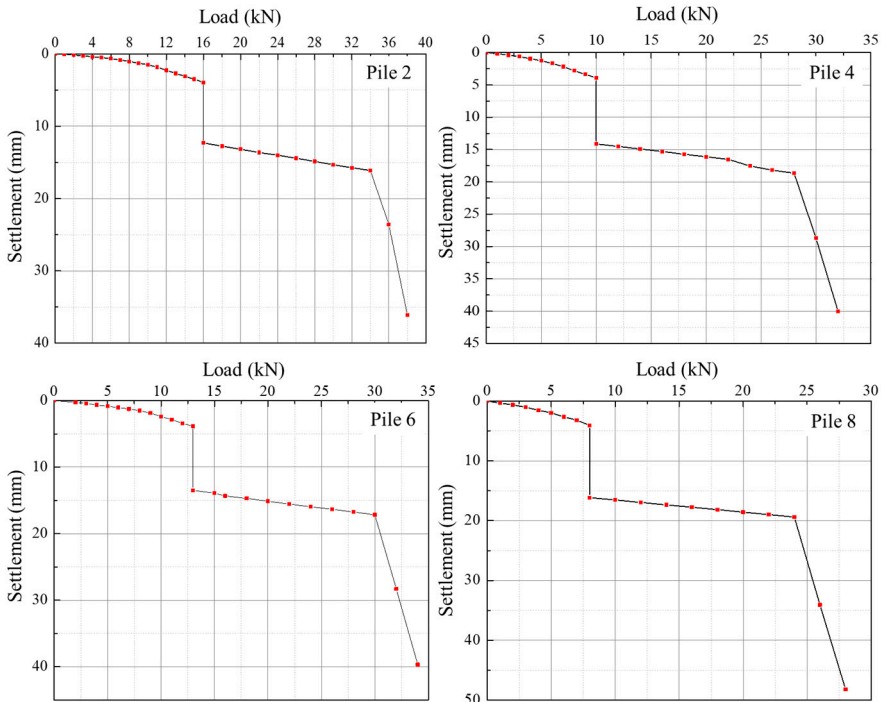

**Figure 11.** Settlement of piles top for piles 2, 4, 6, and 8.

During water immersion, there was a considerable change in the axial force of model piles. As shown in Figure 12, the axial force attenuation of the pile was larger in the water immersion state than in the non-immersion state, with an obvious peak at the lower part of the pile. The maximum axial forces of piles without casing (piles 2 and 4) were even larger than the peak load at the pile top. In addition, the axial force attenuation of the casing section of casing piles (piles 6 and 8) was almost negligible, similar to the non-immersion test.

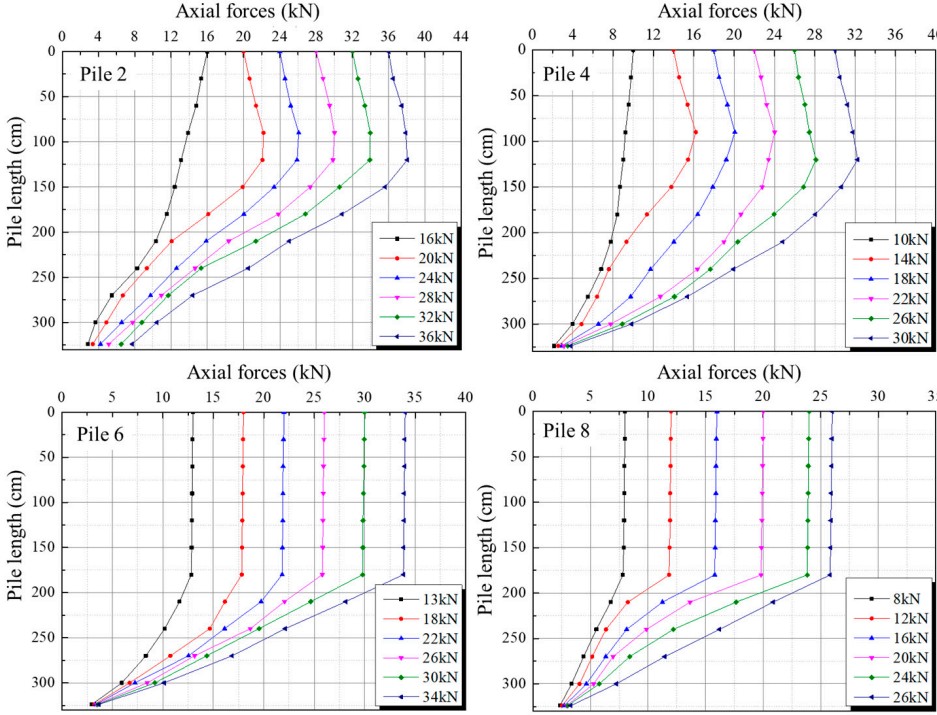

**Figure 12.** Axial force for piles 2, 4, 6, and 8 under immersion condition.

The side frictional resistance along the pile for different loading conditions of piles 2, 4, 6, and 8 is shown in Figure 13. For piles without casing (piles 2 and 4), the negative skin friction initially increased and reached a maximum at around 1500 mm, and then decreased to zero. The neutral point was located at this maximum value. After passing the neutral point, the positive skin friction of pile 2 continued to increase until it reached maximum value, after which it gradually decreased. However, the positive skin friction of pile 4 continued to increase and reached the maximum value at the pile end. The peak values of both negative skin friction and positive skin friction increased with the increasing load. For piles with casing (piles 6 and 8), the impact of negative skin friction generated by collapsible soil layers was negligible due to the presence of the casing. This indicated that the casing effectively counteracted potential negative skin friction effects, thereby minimizing their influence on the overall stability of piles.

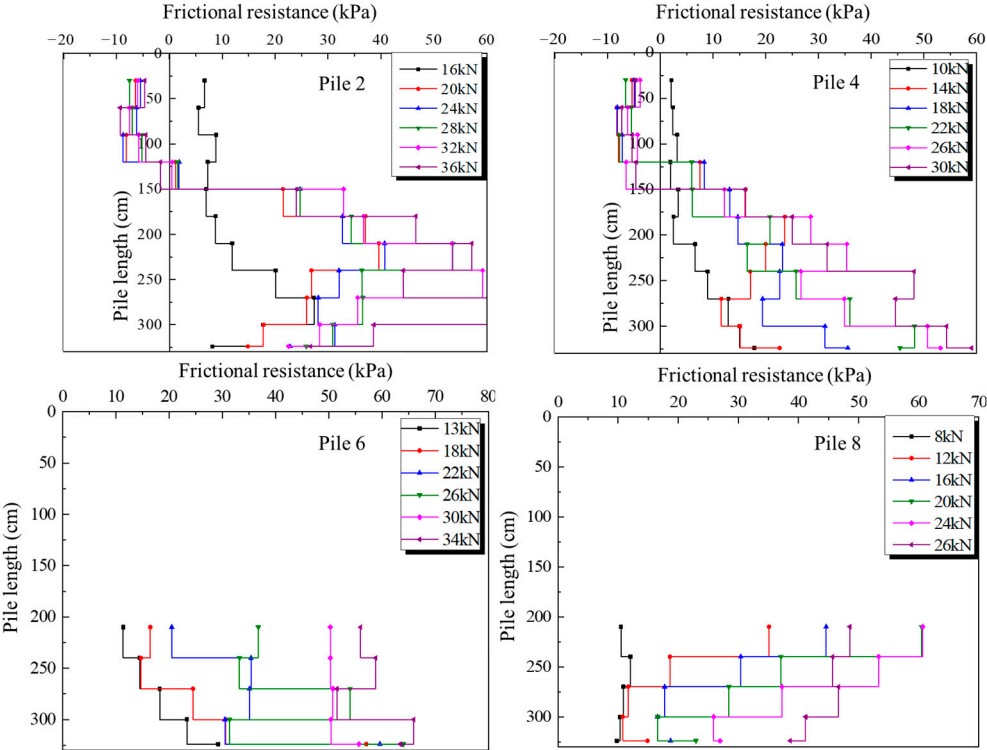

**Figure 13.** Pile side frictional resistance for piles 2, 4, 6, and 8.

## 4. Conclusions

In this study, the settlement, axial force, and side friction resistance of piles in collapsible loess was analyzed based on a series of model experiments. The main conclusions are as follows:

(1) Under the non-immersion condition, the settlement of model piles increased with the increasing pile top load. The ultimate bearing capacity of piles with casing was smaller than piles without casing. The axial force gradually decreased along the pile length for piles without casing. The axial force attenuation of the casing section of casing piles was almost negligible due to the isolating frictional resistance effect of casing.

(2) The settlement of each soil layer increased with the increase in immersion time due to the collapsibility of loess. The settlement process was divided into three stages: initial gradual stage, rapid drop stage, and later gradual stage. The settlement of each soil layer first increased slowly, then rapidly dropped before stabilizing at a steady rate in the later gradual stage.

(3) Under the immersion condition and without additional pile top load, all model piles were in a tension state and the axial force curve exhibited a "D-shape". The positive

and negative skin frictions of piles were alternately distributed in the range of pile length, and increased with the increasing immersion time.

(4) Under the immersion condition and with additional pile top load, the settlement of the pile top increased slowly with increasing load and then entered a stage of significant decrease. The maximum axial force of piles without casing exceeded the peak load at the pile top. The peak values of both negative and positive skin frictions increased with the increasing load.

(5) The presence of steel casing isolated the friction force of loess soil on the upper part of piles, and reduced the failure of the pile foundation in collapsible loess areas.

**Author Contributions:** Q.C.: Resources, Data curation, Visualization; T.C.: Writing—review & editing; Z.L.: Formal analysis, Investigation; D.S.: Software, Writing—original draft, Writing—review & editing; C.W.: Funding acquisition, Validation, Writing—review & editing. All authors have read and agreed to the published version of the manuscript.

**Funding:** This research was funded by [the National Natural Science Foundation of China] grant number [No. 42177141].

**Institutional Review Board Statement:** Not applicable.

**Informed Consent Statement:** Not applicable.

**Data Availability Statement:** Not applicable.

**Conflicts of Interest:** The authors declare no conflict of interest.

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
