# Peer review of "Experimental Study on the Negative Skin Friction of Piles in Collapsible Loess"

_sustainability, doi:10.3390/su15118893_

Round 1

Reviewer 1 Report

The article titled "Experimental study on the negative skin friction of piles in collapsible loess" presents significant results. However, several significant revisions must be addressed before the second round of review.

Firstly, the Abstract must be concise and informative, covering essential aspects such as the study's background, research question, hypothesis, methodology, key findings, and conclusions. Additionally, it should highlight the primary implications and broader context of the research. Secondly, the methodology needs to be strengthened to enable other researchers to replicate the study. If the methodology has been previously published, a brief summary and citation of the source should be provided. Thirdly, the introduction section requires a more detailed discussion leading to the problem statement and scope of the study. Additionally, more relevant literature needs to be discussed, with specific references provided as follows: lines (Collapsible loess is a type of soil that is prone to softening and collapsing when it comes into contact with water):  https://doi.org/10.1007/s13369-022-07565-z; Lines (Additionally, the axial force attenuation in the casing part of the casing piles was almost zero, indicating that the isolating frictional resistance effect of casing was significant.) cite: https://doi.org/10.1139/cgj-2018-0262.

Fourthly, the article's novelty and objectives should be emphasized in the last paragraph of the Introduction. Line (The settlement increased as the immersion time extended, and the sedimentation mode of each soil layer was similar.) require citation. Line (After about 150 h of immersion, the cumulative settlement amount at each soil layer reached almost 70% of the total settlement amount) please explain its implications Lastly, the conclusion needs to provide a comprehensive summary of all the novel and significant findings of the study, along with an explicit statement of the work's significance

Moderate editing of English language

Reviewer 2 Report

In this paper, the authors study the negative skin friction of piles in soft and collapsible loess based on model tests. The settlement, axial force and side friction resistance are investigated in details under both dry and immersion conditions, which can provide guideline for application of piles in collapsible foundation. My comments are listed as follow:

1. For model tests, it would be better to conduct similar design, which is significant to interpret data got from model test into prototype. Unfortunately, I cannot find this part in this work. You can find two papers (10.12989/gae.2022.29.2.133 and 10.1061/(ASCE)CF.1943-5509.0000965) as an example to conduct this part of work.

2. Could you please clearly state the main novelties of this work? I feel a little bit confused.

This work is interesting and worth publishing after major revision.

No.

Round 2

Reviewer 1 Report

Accept in present form

Minor checks required 

Reviewer 2 Report

The authors have revised this paper according to reviewers' comments. It thus can be accepted.

It is readable.